# Surgical and medical management in the treatment of proximal tibial metaphyseal fracture in immature dogs

Carly Sullivan[1], Joshua Zuckerman[2], Daniel James[3], Karl Maritato[4], Emily Morrison[5], Riccarda Schuenemann[6], Ron Ben-Amotz[7]*

1 Small Animal Surgery, BluePearl Veterinary Partners, Levittown, Pennsylvania, United States of America, 2 Small Animal Surgery, Cape Cod Veterinary Specialists, Buzzards Bay, Massachusetts, United States of America, 3 Small Animal Surgery, Small Animal Specialist Hospital, Sydney, Australia, 4 Small Animal Surgery, MedVet Medical & Cancer Center for Pets, Cincinnati, Ohio, United States of America, 5 Veterinary Radiology, MedVet Chicago, Chicago, Illinois, United States of America, 6 Small Animal Surgery, Small Animal Department, Ear, Nose and Throat Unit, College of Veterinary Medicine, University of Leipzig, Leipzig, Germany, 7 Small Animal Orthopedics, Koret School of Veterinary Medicine, Rehovet, Israel

* ron.benamotz@mail.huji.ac.il

**Data Availability Statement:** All relevant data are within the manuscript.

**Funding:** The authors received no specific funding for this work.

## Abstract

The purpose of this study was to report approaches to surgical and medical management of proximal tibial metaphyseal fractures (PTMF) and short-term case outcome. Medical records of immature dogs with PTMF were reviewed and data were collected including history, signalment and side affected. Data pertaining to surgical and medical management including radiographic evaluation and short-term complications were recorded. Forty-five dogs with a total of 47 PTMF identified and treated between 2007–2019 were included in this study. Six cases were managed with external coaptation alone. Forty-one cases were treated surgically with constructs including K-wires in different configurations, bone plate and screws, and external skeletal fixation. Of the cases managed conservatively, 4 developed complications, including bandage sores, diffuse osteopenia of the tarsus/metatarsus, and angular limb deformities. Surgical complications including pin migration necessitating removal, osteopenia, and screw placement in the proximal tibial growth plate or into the stifle joint were found in 16 cases. PTMF treated with surgery had a subjectively more predictable outcome compared to those treated with external coaptation alone. Conservative management may result in complications including development of excessive tibial plateau angle (TPA) as well as distal tibial valgus.

## Introduction

Tibial fractures account for approximately 20% of all long bone fractures in companion animals, making them the third most commonly occurring fracture [1, 2]. Fractures involving the proximal tibial metaphysis are relatively uncommon, and are reported to comprise 3.7% of all tibial fractures [3]. Other fractures of the proximal tibia include tibial tuberosity avulsion fractures, Salter Harris Type II fractures, and combined tibial tuberosity avulsion and proximal physeal fractures [4, 5].

**Competing interests:** The authors have declared that no competing interests exists.

Until recently, it had been suggested that fractures of the proximal tibial metaphysis were exclusive to mature animals secondary to severe trauma [6–8]. However, Deahl et al. reported the occurrence of proximal tibial metaphyseal fractures (PTMF) in juvenile dogs (mean age 4.6 months) following minimal trauma [9]. Most commonly, PTMF manifest in a characteristic curvilinear configuration (Fig 1). The authors of that study suggested that the transition from diaphyseal to metaphyseal bone and the immature or transition zone of the metaphysis play a role in the development of this fracture configuration [9].

In most reported proximal metaphyseal tibia fractures, craniomedial displacement of the distal tibia fragment relative to the proximal fragment occurs. This results in caudolateral angulation of the distal limb. The cranial displacement of the distal fragment and caudal tipping of the proximal tibia increases the risk for development of a steep tibial plateau angle and therefore increased strain on the cranial cruciate ligament as it heals (Fig 2) [10]. In the frontal plane, these fractures may also result in valgus angulation in the distal fragment (Fig 3C).

Clinical outcomes for dogs following management of PTMF have not been reported in the literature. Treatment options for stabilization of these fractures are influenced by multiple factors, including patient age, the presence of open physes, degree of fragment displacement, availability of proximal metaphyseal bone stock, and cost to the client.

The objective of this study was to report on surgical and conservative approaches to the management of PTMF, including complications and short-term outcome.

### Study design

Medical records of dogs that presented with a PTMF between the years of 2015–2020 were collected from several veterinary referral hospitals. Data retrieved from the medical records included breed, weight, side affected, age at the time of presentation, gender, sterilization status at the time of the injury, details of the inciting trauma, displacement, presence of concurrent fibular fracture, stabilization technique (external coaptation vs. surgical fixation) and recorded complications.

## Results

Forty-seven fractures occurring in 45 dogs were available for review (Table 1).

### Signalment

Mean age at the time of the injury was 18.5 weeks (range from 10–39 weeks). Twenty-three cases were female dogs and 20 cases were male dogs. Two cases did not have this data available for review. Sixteen of the cases were intact females and 4 were spayed females. Seventeen cases were intact males and 3 cases were neutered males. In 5 dogs, sterilization status was not recorded. Represented breeds included mixed breed (9), Chihuahua (7), French Bulldog (6), Miniature Poodle (6), Boston Terrier (4), Yorkshire Terrier (3), Corgi (1), Cavalier King Charles Spaniel (1), Maltese (1), Prague Ratter (1), Pomeranian (1), Rat Terrier (1), and Shetland Sheepdog (1). Breed was not recorded in 3 cases. Mean weight was 3.9 kg (range from 0.8–9.9 kg).

### Laterality

The right tibia was affected in 17 cases and the left tibia was affected in 22 cases. Two cases were bilaterally affected. Four cases were lacking information regarding laterality.

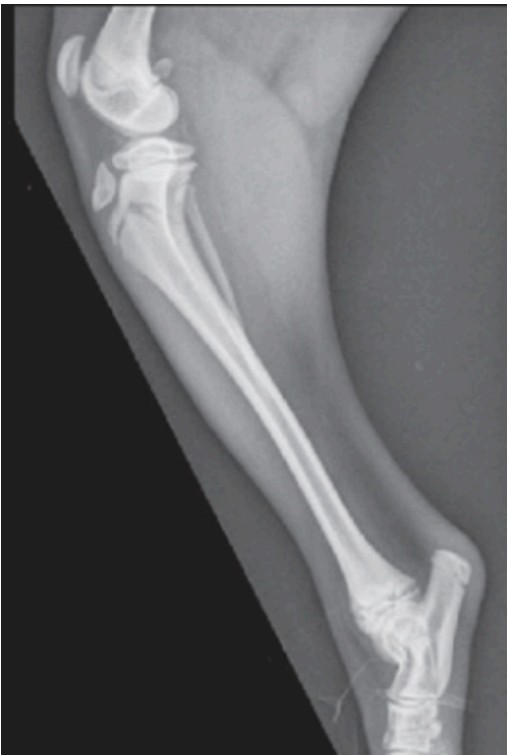

**Fig 1. Characteristic curvilinear configuration of the proximal tibia seen with PTMF.**

### Fracture causes

In all but three cases, the reported cause of the fracture was a fall or jump from a low height, which was consistent with reported causes described in Deahl et al. [9] The medical records for the remaining three cases did not contain information regarding inciting cause.

### Concurrent fibular fracture

Concurrent fibular fracture occurred in 33 of the 46 cases with mediolateral and craniocaudal radiographs available. Of those managed with external coaptation alone, 3 cases had a concurrent fibular fracture and 2 had intact fibulae. Access to only one radiographic view of one medically managed case made it challenging to determine whether a fibular fracture was present. Of the cases managed surgically, 30 sustained fibular fractures and 12 did not have a fibular fracture.

### Fracture treatment

Of the 47 fractures available for review, 6 were treated non-surgically with external coaptation consisting of a bandage and splint for a mean period of 5 weeks (Table 1).

Forty-one fractures were treated surgically. Fracture stabilization using K-wires was performed in 26 cases, 23 of which were repaired with a cross-pinning technique, and 3 of which were repaired with pins and a tension band wire. A soft padded bandage or cranial splint was applied in 9 of these cases for a mean of 17.2 days postoperatively. In 14 cases, stabilization was performed using a bone plate and screws (Table 1). Plates applied included a non-locking T-plate (5), locking T-plate (7) (Fig 4), locking L-plate (1), non-locking L-plate (1), locking TPLO plate (1) (Fig 5), and a dynamic compression plate (1). Two of the plated cases were treated with either a soft padded bandage or lateral splint for a mean of 17.5 days postoperatively.

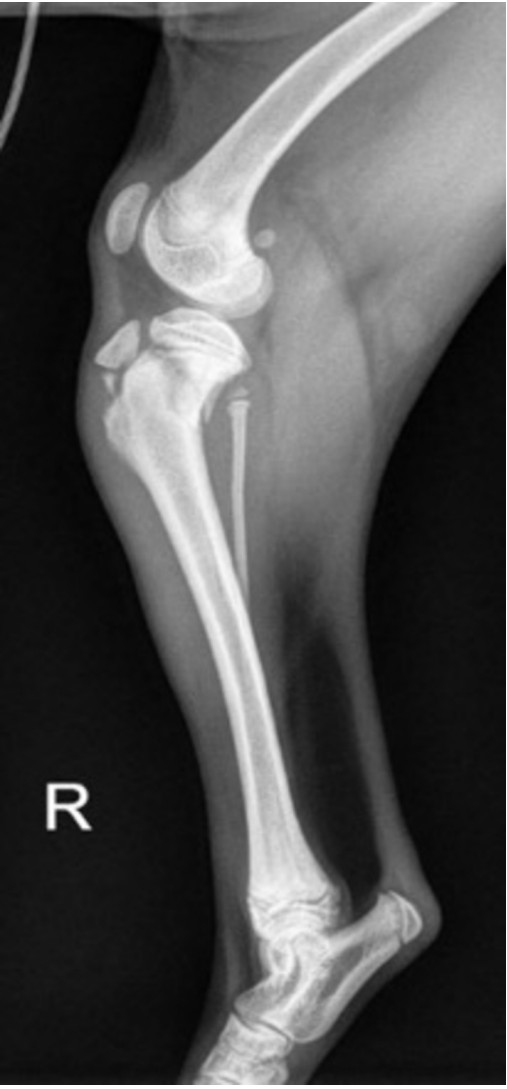

**Fig 2. Mediolateral radiograph of a PTMF demonstrating cranial displacement of the distal fragment and caudal tipping of the proximal tibia resulting in an increased tibial plateau angle.**

One case was treated with an intramedullary pin and modified type 1a external fixator for 8 weeks (Fig 6).

## Complications

Of the 6 cases managed with external coaptation alone, 4 developed complications. Following the development of genu varum, medial patellar luxation, tibial tuberosity avulsion fracture, patella alta, tarsal osteopenia, and fibular malunion after 6 weeks of management in a cranial splint (Fig 3), one patient ultimately underwent amputation of the left pelvic limb (case 34). One bilaterally affected case developed internal tibial rotation and excessive TPA with bilateral medial patellar luxation and diffuse osteopenia of the tarsus and metatarsus. The patellar luxations ultimately required surgical correction (case 40, Fig 7). The remaining case developed bandage sores and disuse osteopenia that resolved following bandage removal (case 45).

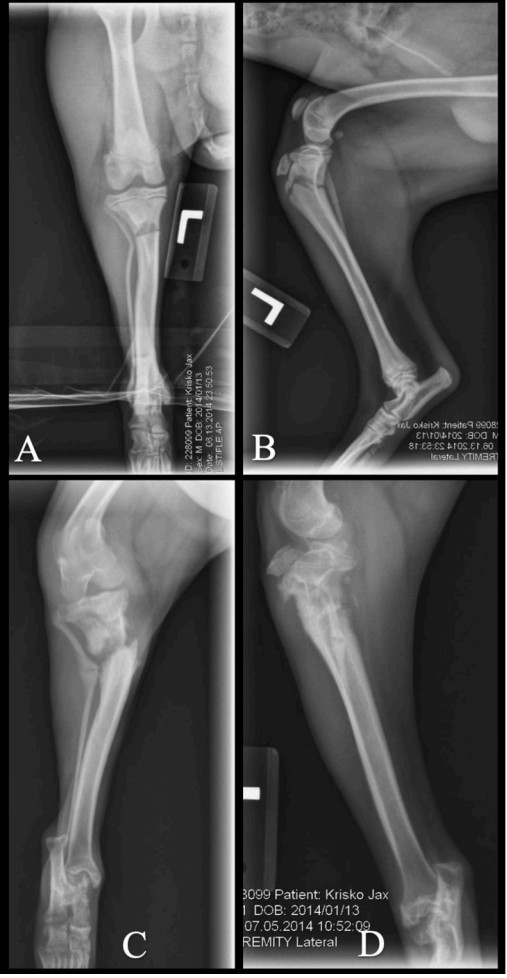

**Fig 3. Case 34 managed with a splint bandage alone.** A and B: Mediolateral and craniocaudal views at time of injury. C and D: Mediolateral and craniocaudal views 3 weeks post injury demonstrating valgus deviation of the distal tibia. Ultimately development of genu varum, medially luxating patella, tibial tuberosity avulsion fracture, patella alta, tarsal osteopenia, and fibular malunion led to an amputation.

Complications were recorded in 9 cases that underwent surgical repair using pins or pins and tension band wire. Pin migration or breakage was the most common complication and occurred in 6 cases (cases 12, 21, 28, 29, 33, and 35) all of which required a second surgery to either replace or remove the displaced implants. In one case (case 37a and b), external coaptation was used as repair augmentation (3 weeks of a cranial splint followed by 1 week of modified Robert Jones bandage). This resulted in development of diffuse osteopenia of the tarsus and metatarsus, which improved following bandage removal (Fig 8). The remaining pin construct case with a complication was continued to exhibit an intermittent lameness at the time of the last follow-up 8 weeks after surgery. In this case it was noted that the patella rode along the medial trochlear ridge but could not be luxated (case 32).

Seven cases that underwent internal fixation using a plate and screws developed complications. The most proximal screw passed through the proximal tibial physis and was left in place (case 6 (Fig 9), 11, and 41 (Fig 10)). This resulted in development of a valgus deformity at the fracture site in 2 cases (case 6 and 11). There was no apparent dysfunction of the limb during follow-up. In two cases, a screw violated the proximal tibial physis and penetrated the stifle

**Table 1. Description of all cases.**

| Case Number | Age at time of injury (weeks) | Gender | Breed | Side | Type of Injury | Displacement | Fibula fracture? | Management | External coaptation | Complications? | Outcome |
|---|---|---|---|---|---|---|---|---|---|---|---|
| 1 | 17 | F | Prague Ratter Dog | Left | Fell from owner's arms | Craniomedial | Yes | 1.2 mm and 0.8 mm IM pins and another 0.8 mm cross pin, TBW with PDS | None | None | Recovered well |
| 2 | 23 | M | Chihuahua | Not Specified | Fell from owner's arms | Cranial | Yes | IM pin and external fixator type I | Only around fixator | None | Recovered well, fixator removed |
| 3 | 14 | M | Chihuahua | Right | Minor trauma during unaccompanied exercise outside—presented 10 days after injury | Cranial | No | 3 k-wires and PDS TBW | 1 week SPB | None | Recovered well |
| 4 | 13 | F | Mix | Left | Jumped off roof | Cranial | Yes | 3 1.2 mm k-wires and PDS TBW | MRJ bandage recommended but patient lost to follow-up | Lost to follow-up | Lost to follow-up |
| 5 | 22 | M | Miniature Poodle | Not Specified | Jumped out of owner's arms 3.5 weeks prior to presentation | Cranial | Yes | Not specified | Not specified | Not specified | No surgery—mostly healed at time of consultation but surgery was recommended |
| 6 | 15 | M | Terrier Mix | Right | Dropped from low height | Craniolateral | Yes | ORIF with non-locking T-plate | None | Valgus deformity—screw impingement on lateral portion of proximal tibial physis | ? |
| 7 | 13 | M | Boston Terrier | Left | Dropped from low height | No preoperative radiographs to evaluate | Yes | ORIF with non-locking T-plate | None | None | Recovered well |
| 8 | 18 | M | Toy Poodle | Left | Dropped from low height | Craniomedial | Yes | ORIF with locking T-plate | None | Valgus deviation at the fracture site | ? |
| 9 | 10 | FS | Poodle x CKCS | Left | Fell while running | Cranial | No | ORIF with crossed k-wire fixation | None | None | Recovered well |
| 10 | 11 | F | Chihuahua | Right | Dog fight with history of being dropped 1 week prior to fight | Medial | Yes | ORIF with crossed k-wires and non-locking L-plate fixation | None | Discharging sinus without lameness 3 months later resulted in explantation | Recovered well after explantation |
| 11 | 16 | F | Pomeranian | Left | Unknown trauma | Craniomedial | Yes | ORIF with L locking plate | None | Most proximal screws in proximolateral tibial physis | ? |
| 12 | 14 | F | Toy Poodle | Left | Dropped from low height | Cranial | No | ORIF with k-wire fixation | None | Explantation after fracture healed | Recovered well after explantation |

*(Continued)*

**Table 1.** (Continued)

| Case Number | Age at time of injury (weeks) | Gender | Breed | Side | Type of Injury | Displacement | Fibula fracture? | Management | External coaptation | Complications? | Outcome |
|---|---|---|---|---|---|---|---|---|---|---|---|
| 13 | 13 | F | CKCS | Right | Fell from couch | Craniolateral | Yes | ORIF with non-locking T-plate | Not specified | None noted | Recovered well |
| 14 | 18 | F | French Bulldog | Not Specified | Unknown trauma | Unable to assess on radiographs | No | ORIF with non-locking T-plate | Not specified | External rotation and valgus on radiographs but ambulates well | Recovered well |
| 15 | 20 | F | Toy Poodle | Left | Jumped from sleeper | None | Yes | Mini non-locking T-plate after splinted by pDVM for 2 weeks | None | Valgus deforming but functional, proximal screw into the joint and through proximolateral physis | ? |
| 16 | Not specified | Not specified | Not specified | Left | Not specified | Cranial | Yes | ORIF with k-wire fixation | None | Not specified | Recovered well |
| 17 | 24 | MN | Not specified | Left | Jumped off bed | None | No | 2.7 mm locking T plate | Not specified | Disuse osteopenia of tarsus, proximal screw into joint, physeal violation with screw with increased TPA | Recovered well |
| 18 | 20 | M | Rat Terrier | Right | Unknown trauma at home | None | Yes | Cross pins | None | None | Recovered well |
| 19 | 20 | F | Sheltie | Left | Fell off couch | Only PO radiographs available | No | Cross pins | None | None | Recovered well |
| 20 | 13 | F | Boston Terrier | Left | Unknown trauma at home | Unable to assess on radiographs | No | Cross pins | None | None | Recovered well |
| 21 | | M | Chihuahua Mix | Left | Unknown trauma at home | Only PO radiographs available | Yes | Cross pins | None | Pin migration | Recovered well after explantation |
| 22 | 18 | F | Boston Terrier | Left | Unknown trauma outside in yard alone | None | No | ORIF with locking T-plate | None | None | Recovered well |
| 23 | 19 | M | French Bulldog | Right | Fell down the stairs | Only PO radiographs available | No | IM pin with antirotational pin | None | None | Recovered well |
| 24 | 16 | F | French Bulldog | Right | Fell from steps | Only PO radiographs available | No | PAX T-plate | None | None | Recovered well |
| 25 | 24 | M | Yorkie | Right | Fell from arms | Caudal | Yes | 3 cross pins | None | None | Recovered well |
| 26 | 24 | F | Chihuahua | Right | Unknown trauma at home | Caudal | No | 3 cross pins | None | None | Recovered well |
| 27 | 17 | F | Toy Poodle | Right | Fell off deck | Only PO radiographs available | Yes | PAX T-plate | None | Infection post-operatively | Recovered well after treatment for infection |

(Continued)

**Table 1.** (Continued)

| Case Number | Age at time of injury (weeks) | Gender | Breed | Side | Type of Injury | Displacement | Fibula fracture? | Management | External coaptation | Complications? | Outcome |
|---|---|---|---|---|---|---|---|---|---|---|---|
| 28 | 39 | MN | Maltese | Right | Fell from arms | Caudolateral | Yes | 3 cross pins | None | Pin migration | Recovered well after explantation |
| 29 | 20 | FS | Yorkie | Left | Unknown trauma at home | Medial | Yes | 3 cross pins | None | Pin migration | Recovered well after explantation |
| 30 | 20 | FS | French Bulldog | Right | Unknown trauma at home | Only PO radiographs available | Yes | 3 cross pins | None | None | Recovered well |
| 31 | 19 | F | French Bulldog | Right | Fell off couch | Cranial | Yes | 3 cross pins | None | None | Recovered well |
| 32 | 18 | F | Boston Terrier | Right | Fell from bed | Craniomedial | Yes | IM pin | Cranial splint for 3 weeks, SPB for 1 week | Patella riding medially but unable to luxate | Occasionally lame at home |
| 33 | 24 | M | French Bulldog | Right | Fell from couch | Craniomedial | Yes | 2 cross pins, 4 pins including 1 down tibial shaft, MPL correction | 2 weeks for 1st surgery (pins in joint), 4 weeks for second surgery | Fractured 2 weeks post first surgery with 2 k-wires—repeated surgery with 4 pins at divergent angles and bandaged for 4 weeks. 6 months post 1st surgery, grade 4/4 MPL corrected, 8 months post 1st surgery, pin removal for TTT | Recovered well after last procedure |
| 34 | 20 | M | Poodle/Terrier Mix | Left | Jumped out of stopped car | Cranial | Yes | Cranial splint for 6 weeks | Cranial splint for 6 weeks | Non-healing malunion developing angular limb deformity (genu varum), MPL, tibia tuberosity avulsion, patella alta, tarsal osteopenia, fibular malunion | Left mid-femoral amputation |
| 35 | 20 | M | Yorkie | Left | Fell down the stairs | Cranial | Yes | 2 0.045 k-wires | Cranial splint for 2 weeks | Pin migration | Recovered well after explantation |
| 36 | 24 | F | Terrier Mix | Left | Stepped on by owner | Cranial | Yes | 3 0.045 k-wires | None | None | Recovered well |
| 37a | 18 | M | Chihuahua | Bilateral—Right | Jump off couch | Only PO radiographs available | Yes | 2 0.045 k-wires | Cranial splint for 3 weeks, SPB for 1 weeks | Diffuse osteopenia of the tarsus and metatarsus | Recovered well once external coaptation was removed |
| 37b | 18 | M | Chihuahua | Bilateral—Left | Jump off couch | Only PO radiographs available | Yes | 2 0.045 k-wires | Cranial splint for 3 weeks, SPB for 1 weeks | Diffuse osteopenia of the tarsus and metatarsus | Recovered well once external coaptation was removed |
| 38 | 16 | M | Corgi | Right | Jump off bench | Caudal | Yes | 2 1/16 pins | SPB overnight | None reported | Lost to follow-up |
| 39 | 16 | F | Chihuahua | Right | Fell from bed | None | No | 2 0.045 k-wires | SPB for 1 week | None reported | Lost to follow-up |

(*Continued*)

**Table 1.** (Continued)

| Case Number | Age at time of injury (weeks) | Gender | Breed | Side | Type of Injury | Displacement | Fibula fracture? | Management | External coaptation | Complications? | Outcome |
|---|---|---|---|---|---|---|---|---|---|---|---|
| 40a | 20 | M | Chihuahua Mix | Bilateral—Left | Jump from owners' arms | Cranial | No | Lateral splint for 2 weeks, SPB for 2 weeks | Lateral splint for 2 weeks, SPB for 2 weeks | Internal tibial rotation, diffuse osteopenia of tarsus and metatarsus, changes to the metatarsus | Grade 1-2/4 MPL |
| 40b | 20 | M | Chihuahua Mix | Bilateral—Right | Jump from owners' arms | Craniolateral | Yes | Lateral splint for 3 weeks, SPB for 3 weeks | Lateral splint for 3 weeks, SPB for 3 weeks | Internal tibial rotation, diffuse osteopenia of tarsus and metatarsus, changes to the metatarsus | Patella rides on medial trochlear ridge but unable to luxate |
| 41 | 16 | MN | Chihuahua Mix | Left | Jumped from owners' arms | Caudal | Yes | 0.045 k-wires, 2.0 DCP plate | Splint for 2 weeks, SPB for 2 weeks | Screw through proximolateral tibia physis on immediate PO radiographs but not at recheck | Recovered well |
| 42 | 24 | F | Chihuahua | Not specified | Not specified | Only post treatment radiographs available | Unsure | Splint for 4 weeks | Splint for 4 weeks | None | Recovered well |
| 43 | Not specified | Not specified | Not specified | Left | Not specified | None | Yes | 3 cross pins | None | None | Recovered well |
| 44 | 16 | M | Terrier Mix | Left | Jumped from bed | Caudomedial | Yes | TPLO plate | SPB for 1 week | None | Recovered well |
| 45 | 13 | FS | Minatare Poodle | Left | Jumped from bed | Craniomedial | No | Splint for 4w - changed weekly | Splint for 4w - changed weekly | Mild skin sores, valgus, disuse osteopenia | Recovered well |

IM–intramedullary

TBW–tension band wire

PDS–polydioxanone

K-wire–Kirschner wire

SPB–soft padded bandage

MRJ–modified Robert Jones

ORIF–open reduction, internal fixation

MPL–medial patellar luxation

TTT–tibial tuberosity transposition

PO—postoperative

TPLO–tibial plateau leveling osteotomy

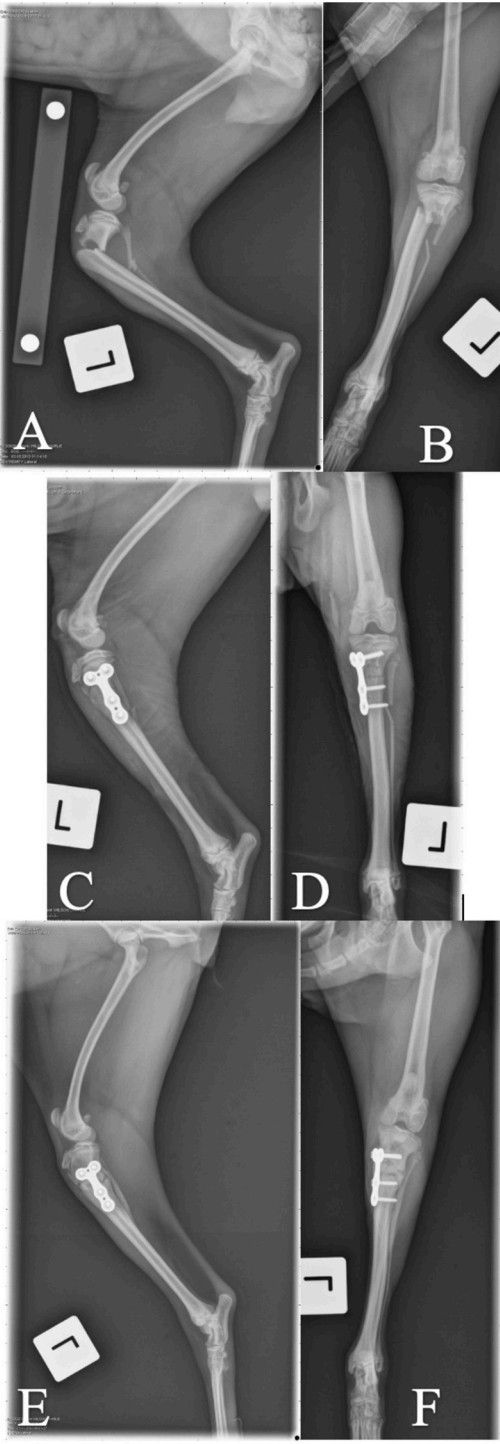

**Fig 4. Case 8 managed with locking T-plate for stabilization of PTMF.** A and B: Preoperative mediolateral and craniocaudal views. C and D: Immediate postoperative mediolateral and craniocaudal views. E and F: 6 weeks postoperative mediolateral and craniocaudal views.

joint (case 15 and 17, Fig 11). One case was lost to follow up and the other recovered well without any reported lameness. The remaining two cases treated with plate and screw fixation

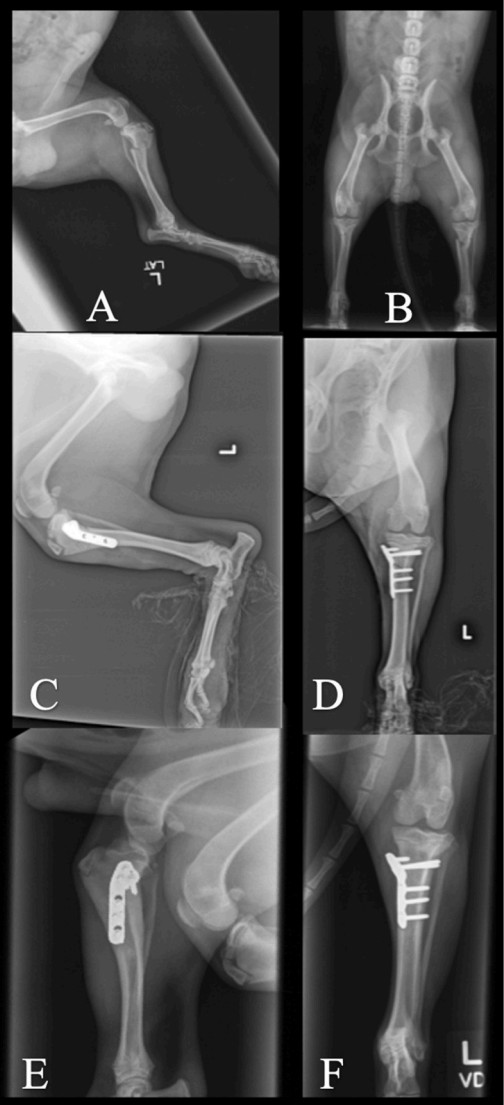

**Fig 5. Case 47 managed using a locking TPLO plate.** A and B: Mediolateral and craniocaudal views. C and D: Immediate postoperative mediolateral and craniocaudal views. E and F: 4 weeks postoperative mediolateral and craniocaudal views.

developed surgical site infections that resolved with either implant removal 12 weeks after surgery or oral antibiotic therapy (case 10 and 27, respectively).

## Discussion

Both surgical and conservative approaches to the management of proximal tibial metaphyseal fractures have been described. Clinically, patients treated with surgical stabilization appeared to have superior outcomes compared to those treated with external coaptation alone. However, the small sample size of the present study precludes the demonstration of statistical significance.

Despite the small sample site, the authors believe there is evidence to suggest that surgical stabilization of these cases achieves better clinical outcome with a lower risk of severe

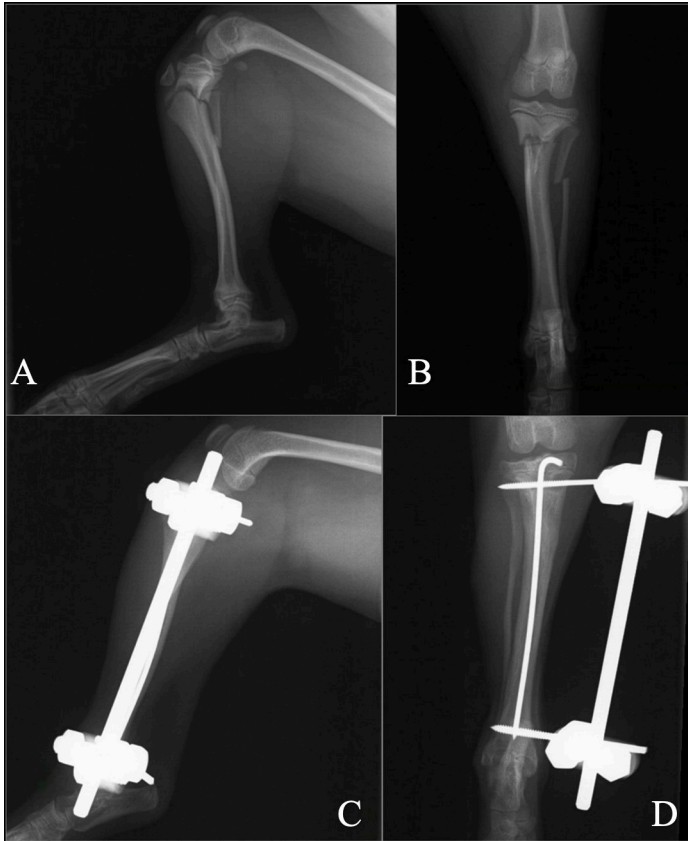

**Fig 6. Intramedullary pin and modified type 1a external fixator (Case 2).** A and B: Preoperative mediolateral and craniocaudal views. C and D: 8 weeks postoperative radiographs: mediolateral and craniocaudal views.

complications compared to use of external coaptation alone. Fracture of the proximal tibial metaphysis typically occurs in juvenile dogs [6], and the use of external coaptation during periods of sustained growth may result in complications including angular limb deformity, muscle contracture, and/or disuse osteopenia [11]. Additionally, bandages may also contribute to the development of sores, swelling, and dermatitis [12]. One patient in this study required amputation of the affected limb as a direct result of complications arising from external coaptation. Given the potential for significant complications, the authors believe external coaptation alone should only be considered for cases in which limited financial resources make surgical correction impossible.

Of the fractures that underwent primary surgical repair, 26 were stabilized using a pin construct, 14 were stabilized using a bone plate and screws and one was stabilized with a type 1a external fixator. Regardless of stabilization method, all surgical cases had a more predictable outcome when compared to the cases managed with external coaptation alone.

Internal fixation constructs using pins included cross pinning, multiple diverging K-wires and intramedullary pin placement. The most common complication encountered in these cases was pin migration, often necessitating removal following documentation of adequate fracture healing. Three previous studies on fracture pinning showed a variable pin removal rate after reduction of physeal fractures. Boekhout-Ta et al. reported a pin migration rate of 4% and elective pin removal due to irritation was performed in 41% of cases in this study [13]. In 2004, it was reported that no pins migrated or were removed in 7 young dogs undergoing

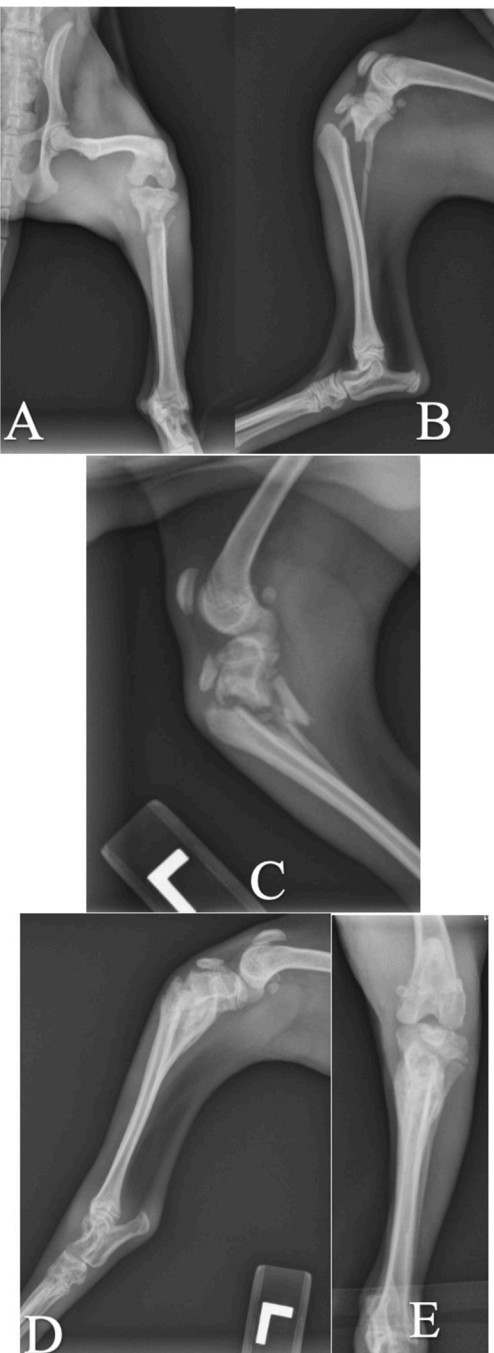

**Fig 7. Case 40a managed with a splint bandage alone.** A and B: Mediolateral and craniocaudal views at time of injury. C: Mediolateral view 2 weeks post injury. D and E: Mediolateral and craniocaudal views 6 weeks post injury demonstrating excessive TPA that can result from treatment with external coaptation alone.

open reduction and internal fixation of proximal tibial fractures [14]. Another study in 1989 evaluated blind pinning of the tibia and femur in dogs. There was a 71% pin removal rate reported in 7 physeal fractures [15]. At this time, more information is needed regarding pin migration rate and removal for cases of PTMF, although removal is considered standard of care if migration or irritation occurs.

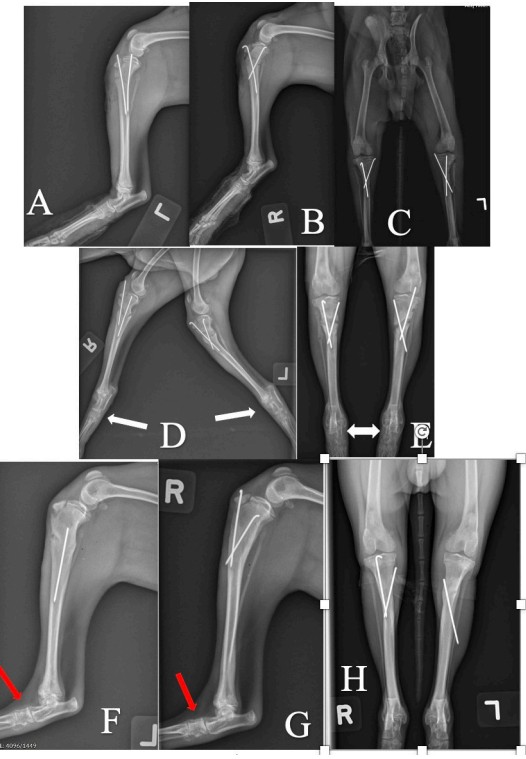

**Fig 8. Development of diffuse osteopenia of the tarsus and metatarsus after a bandage was placed for 4 weeks postoperatively (case 37).** A, B and C: Bilateral mediolateral and craniocaudal views immediately postoperative. D and E: Mediolateral and craniocaudal views 4 weeks postoperative. There is progressive healing of the proximal tibial fractures. Severe osteopenia is present affecting the tarsal cuboidal bones and proximal metatarsal bones (white arrows). F, G and H: Mediolateral and craniocaudal views 8 weeks postoperative. The proximal tibial fractures have healed appropriately. Mild to moderate osteopenia of distal limbs, but improved compared to radiographs at 4 weeks after surgery (white arrows).

Subjectively, we appreciated an increased difficulty associated with reduction of the fracture site during surgery when pins were used. Interestingly, even in dogs that underwent primary repair with pins, bandages appeared to increase the risk of complications, though the significance of this finding could not be demonstrated due to low case numbers. However, the use of external coaptation following repair of PTMF with pins should be approached cautiously. It is recommended to closely follow up on cases that were repaired with pins both clinically and radiographically in order to address any complications such as pin migration and soft tissue mobility.

The most common complication encountered in the cases managed with a bone plate and screws was inadvertent placement of the most proximal screw through the proximal tibial physis. This resulted in valgus deviation of the proximal tibia in 2 cases (Fig 9), but did not affect the clinical outcome of these patients in the short-term. Long-term follow up would be necessary to fully determine whether this change is clinically significant. One case with physeal violation continued grow normally and did not develop distal tibial valgus (case 41, Fig 10). Kennon et al. described that limited central transphyseal bridging that occurs after physeal injury can be associated with continued normal hydrostatic bone growth to overcome physeal violation, which may explain the outcome of this case [16]. In two cases, the proximal screw violated the proximal tibial physis and penetrated the stifle joint. It is likely that the use of intraoperative fluoroscopy would greatly diminish the risk of this complication. There is no

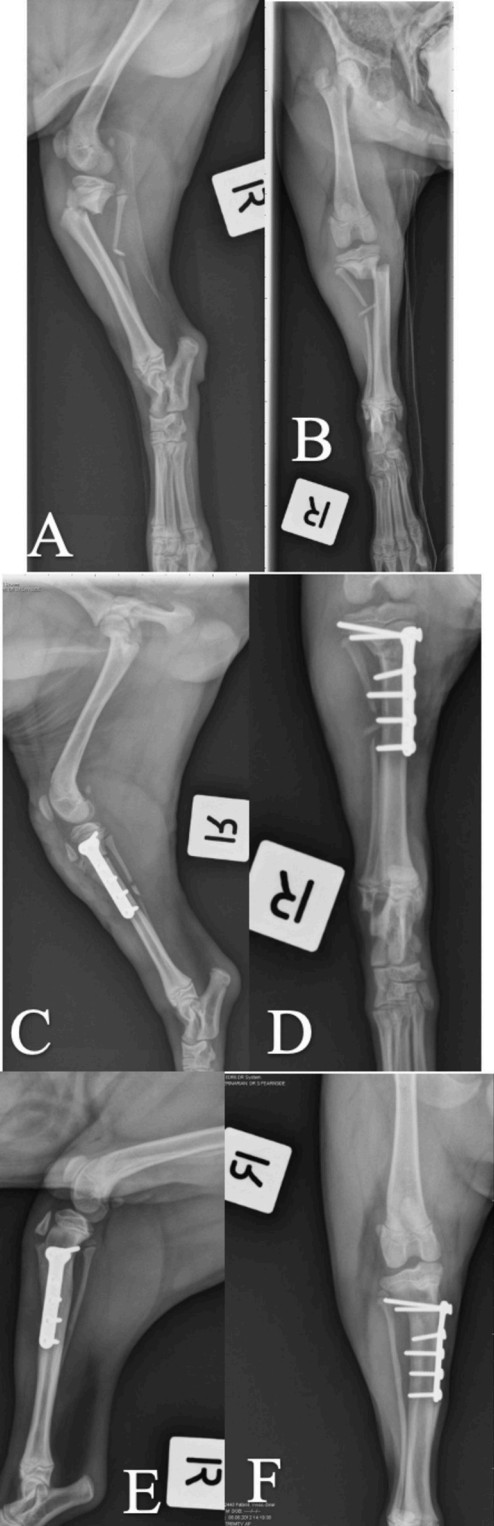

**Fig 9. Inadvertent placement of the most proximal screw through the proximal tibial physis resulting in valgus deviation of the proximal tibia (Case 6).** A and B: Mediolateral and craniocaudal views at time of fracture diagnosis. C and D: Immediate postoperative mediolateral and craniocaudal views. E and F: 6 weeks postoperative mediolateral and craniocaudal views.

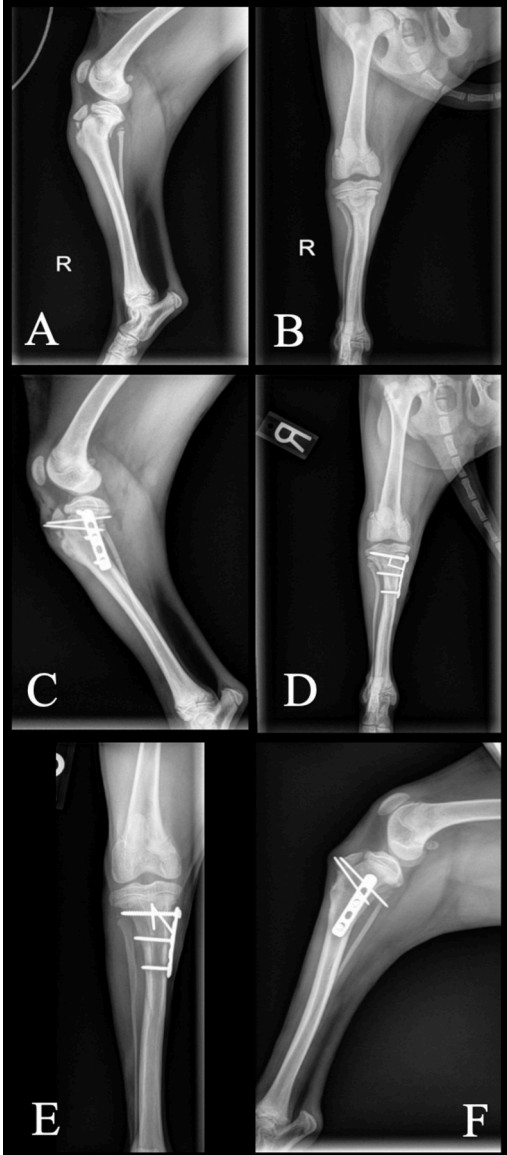

**Fig 10. Case 41 in which the proximal screw violated the proximal tibial physis immediately postoperatively on radiographs, but was not within the proximal tibial physis at radiographic evaluation 4 weeks after surgery.** A and B: Mediolateral and craniocaudal views at time of injury. C and D: Immediate postoperative mediolateral and craniocaudal views. E and F: 4 weeks postoperative radiographs mediolateral and craniocaudal views.

indication in the medical records as to why these cases were not immediately re-operated to achieve appropriate screw placement. Possible reasons for this may be due to limited bone stock in these small patients, the potential for destabilization of the construct, or planned future removal of the implants once the fracture healed. When fluoroscopy is not available, particular care should be taken to evaluate the placement of the proximal screw on postoperative radiographs. If violation of the proximal tibial physis or the articular surface is apparent, revision should be undertaken prior to recovering the patient.

Intraarticular screw placement causes chondral damage, exacerbates chondrolysis and osteoarthritis, and therefore should be avoided [17]. In the two cases with penetration of the proximal screw into the stifle joint, both patients recovered well and did not have any clinical

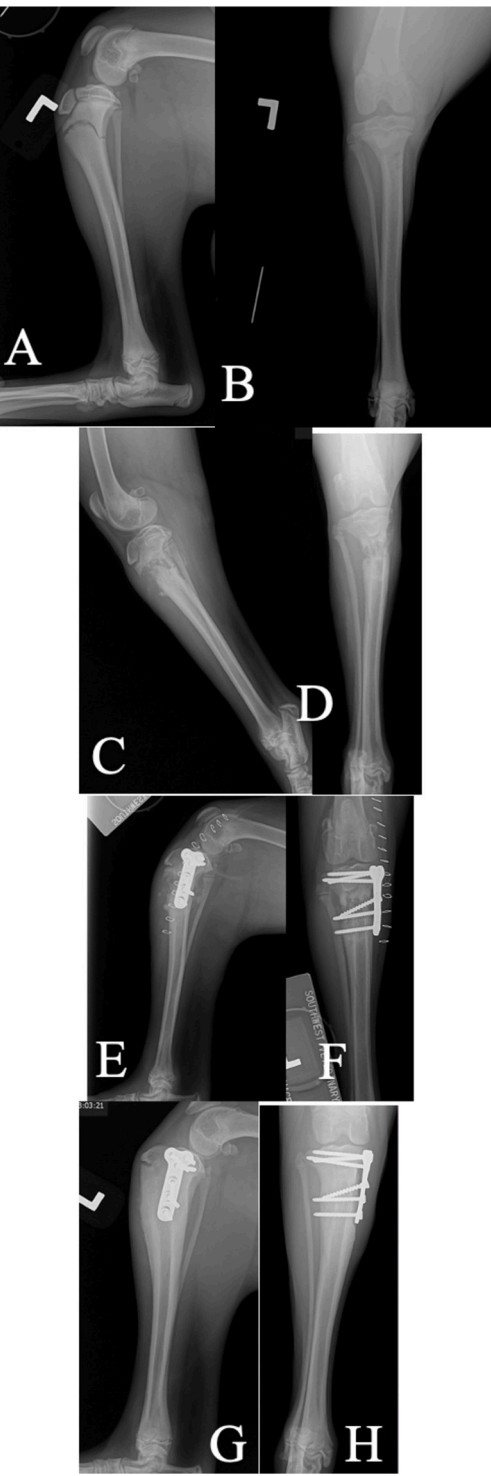

**Fig 11. Inadvertent place of the most proximal screw through the proximal tibial physis and into stifle joint after the case was first medically managed with a splint bandage (Case 17).** A and B: Mediolateral and craniocaudal views performed 1 day after the initial injury: C and D: Mediolateral and craniocaudal views performed 3 weeks post splint placement. There was concern for collapsing of the lateral tibial cortex resulting in increased tibial angulation. The patient was still grade 4/4 lame on exam, so surgical stabilization was elected. E and F: Mediolateral and craniocaudal views immediately postoperatively showing screw placement into the joint. G and H: Mediolateral and craniocaudal views performed 6 weeks postoperatively with screw placement still within the joint, but patient was not lame or painful on exam.

lameness at the time of last follow-up. It is possible that the lack of apparent pain or lameness in these cases is due to skiving, which is defined in human literature as the condition when the subchondral plate is disrupted while the underlying cartilage is physically displaced without the screw entering the joint [18]. CT scan has a greater sensitivity compared to radiography when diagnosing skiving, which was not performed in any of the cases in this study [18]. It is possible that these cases had skiving as opposed to intraarticular screw violation, which accounts for the favorable outcome in both patients.

One case was managed with an intramedullary pin and type 1a external fixator (Fig 5) and recovered well without any noted complications. This type of fracture fixation method may be useful for PTMF given the limited bone stock of the proximal fracture fragment, its proximity to the proximal tibial physis and the potential for rapid healing due to patient age [19].

Of particular concern in canine PTMF is the cranial displacement of the distal segment, which results in caudal tipping of the proximal tibia and risks the development of an excessive tibial plateau angle, potentially increasing strain on the cranial cruciate ligament (Figs 2 and 7) [10]. Surgical intervention allows for more accurate reduction of the fracture fragments, therefore reducing the risk of caudal tipping of the proximal segment. None of the patients in this study developed cranial cruciate ligament rupture, but this may be attributed to the short-term follow-up. We also recognized that PTMF configuration may result in distal tibial valgus. This may also be corrected by the superior reduction and alignment afforded by surgical intervention as opposed to medical management with external coaptation.

Limitations of this study include those inherent to its retrospective nature, a low number of cases that precluded statistical analysis between and within the various groups, inconsistent methods of surgical stabilization employed, inconsistent use of postoperative bandages and splints, and a lack of consistency of radiographic evaluation among the cases reviewed. Further studies are needed to determine the most effective method of surgical intervention for fracture repair.

Despite limitations within this study, we reported surgical and medical management of PTMF fractures along with the short-term outcome. Subjectively, surgical management has a more predictable outcome and can prevent conformational changes to the proximal tibia that may predispose patients to cranial cruciate ligament rupture and angular limb deformity.

## Acknowledgments

The authors acknowledge Rebecca Ball, Steven Martin, Kate Fitzwater, Melissa Hobday, Jennifer MacLeod, Robert Orsher, Nicole Salas, Rachel Seibert and Kellie Stirling for their contribution of cases for this study.

## Author Contributions

**Conceptualization:** Carly Sullivan, Joshua Zuckerman, Daniel James, Karl Maritato, Emily Morrison, Riccarda Schuenemann, Ron Ben-Amotz.

**Data curation:** Carly Sullivan, Daniel James, Karl Maritato, Emily Morrison, Riccarda Schuenemann, Ron Ben-Amotz.

**Formal analysis:** Carly Sullivan, Joshua Zuckerman, Daniel James, Emily Morrison, Riccarda Schuenemann, Ron Ben-Amotz.

**Investigation:** Carly Sullivan, Joshua Zuckerman, Daniel James, Karl Maritato, Emily Morrison, Riccarda Schuenemann, Ron Ben-Amotz.

**Methodology:** Carly Sullivan, Joshua Zuckerman, Daniel James, Karl Maritato, Emily Morrison, Riccarda Schuenemann, Ron Ben-Amotz.

**Project administration:** Carly Sullivan, Joshua Zuckerman, Daniel James, Emily Morrison, Riccarda Schuenemann, Ron Ben-Amotz.

**Resources:** Carly Sullivan, Joshua Zuckerman, Daniel James, Karl Maritato, Ron Ben-Amotz.

**Software:** Carly Sullivan.

**Supervision:** Joshua Zuckerman, Daniel James, Karl Maritato, Riccarda Schuenemann, Ron Ben-Amotz.

**Validation:** Carly Sullivan, Joshua Zuckerman, Daniel James, Karl Maritato, Emily Morrison, Riccarda Schuenemann, Ron Ben-Amotz.

**Visualization:** Carly Sullivan, Joshua Zuckerman, Daniel James, Karl Maritato, Emily Morrison, Riccarda Schuenemann, Ron Ben-Amotz.

**Writing – original draft:** Carly Sullivan, Joshua Zuckerman, Daniel James, Karl Maritato, Emily Morrison, Riccarda Schuenemann, Ron Ben-Amotz.

**Writing – review & editing:** Carly Sullivan, Joshua Zuckerman, Daniel James, Karl Maritato, Emily Morrison, Riccarda Schuenemann, Ron Ben-Amotz.

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
