## [Decision Letter · Decision Letter 0]

17 Feb 2022

PONE-D-21-29374Surgical and medical management in the treatment of proximal tibial metaphyseal fracture in immature dogsPLOS ONE

Dear Dr. Ben-Amotz,

Thank you for submitting your manuscript to PLOS ONE. After careful consideration, we feel that it has merit but does not fully meet PLOS ONE’s publication criteria as it currently stands. Therefore, we invite you to submit a revised version of the manuscript that addresses the points raised during the review process.

PLOS ONE's staff editors have left a comment in the section '**Journal Requirements**' related to the journal's publication criteria. Could you please revise the manuscript to carefully address the concerns raised?

We look forward to receiving your revised manuscript.

Kind regards,

Jason Organ

Academic Editor

PLOS ONE

**Journal Requirements:**

PLOS ONE's publication criteria requires that methodology is described in enough detail such that another researcher could reproduce this research (https://journals.plos.org/plosone/s/criteria-for-publication#loc-3). In your Methods section, please provide additional details regarding the sources of the data used in your study (e.g. the sources of medical records). If there are issues with reporting this such as confidentiality/privacy please provide an explanation for this in your 'data availability' statement, and update the methods section such that search methodology for medical records is described in a manner that could be reproducible by another researcher (for example, the types of records, type of facilities contacted, broad region in which they are located etc.). For more information regarding PLOS' policy on materials sharing and reporting, see https://journals.plos.org/plosone/s/materials-and-software-sharing#loc-sharing-materials.

PLOS ONE's publication criteria includes data availability (https://journals.plos.org/plosone/s/criteria-for-publication#loc-7), and this policy includes requirements (https://journals.plos.org/plosone/s/data-availability) that authors make all data necessary to replicate their study's findings publicly availably without restriction at the time of publication. This policy page provides clarification of specific minimal requirements for data, e.g. data underlying measures of central tendancy/means/aggregates reported in a manuscript. When specific legal or ethical restrictions prohibit public sharing of the data set, authors must indicate how others may obtain access to the data. Please either make the underlying data available (e.g. by uploading with the manuscript or to a repository), or provide clarification on this in your data availability statement. In the section of your submission asking 'Describe where the data may be found in full sentences' the response currently reads 'N/A'. Please provide a response in this section when you resubmit to clarify how the data availability has been addressed.

Please also consider any reviewers comments/feedback in making any corrections to your manuscript.

Additional Editor Comments (if provided):

Reviewers' comments:

Reviewer's Responses to Questions

**Comments to the Author**

1. Is the manuscript technically sound, and do the data support the conclusions?

Reviewer #1: Yes

2. Has the statistical analysis been performed appropriately and rigorously? 

Reviewer #1: N/A

3. Have the authors made all data underlying the findings in their manuscript fully available?

Reviewer #1: Yes

4. Is the manuscript presented in an intelligible fashion and written in standard English?

Reviewer #1: Yes

5. Review Comments to the Author

Reviewer #1: This is a very nice report describing the outcomes of proximal tibia fractures in dogs. The retrospective review of 45 cases, from multiple groups, provides a diverse overview that spans ages, breeds, and approaches to management. The discussion nicely integrates the findings into previous work. This paper will serve as a nice resource for providers who are dealing with these cases and those who are looking to expand on this research with case-controlled studies.

One suggestion, which is not required, is to incorporate some aspect of 'time' into the assessment. That is, it would be beneficial for the reader to have some sense of the duration of the various interventions, the time between various interventions and/or the time at which the fracture was considered 'healed'.

6. PLOS authors have the option to publish the peer review history of their article (what does this mean?). If published, this will include your full peer review and any attached files.

Reviewer #1: No

---

## [Author Response · Author response to Decision Letter 0]

13 Apr 2022

To the Editors of PLOS ONE and Reviewer of “Surgical and medical management in the treatment of proximal tibial metaphyseal fracture in immature dogs”:

Thank you very much for your consideration of this research article. Below is an explanation of changes made in order to appropriately satisfy the PLOS ONE publication criteria:

Thank you for your comment regarding the description in the Methods section. We have provided additional details regarding the data sources used and these changes are noted within the manuscript. The change in the dates reflects when the cases initially presented to the hospitals. We have also included information regarding the number of hospitals from which data was sourced as well as their respective locations. 

We have appropriately addressed PLOS ONE’s publication criteria regarding data availability within the portal. At the time of the initial submission, a misunderstanding prompted the “N/A” response. This has now been resolved. 

One of the references was accidentally duplicated in the reference list twice, and this has now been corrected in the manuscript. One of the references was incorrectly cited in the introduction, so this was corrected. 

Thank you for the reviewer comment: “One suggestion, which is not required, is to incorporate some aspect of 'time' into the assessment. That is, it would be beneficial for the reader to have some sense of the duration of the various interventions, the time between various interventions and/or the time at which the fracture was considered 'healed'”. Unfortunately, the timing data requested is not available for all of the cases we included in this study. The authors were concerned that the inclusion of incomplete data would detract from the study and potentially confuse the readers, particularly since multiple treatment modalities were described. 

Please let us know if any further changes are needed to meet the criteria of PLOS ONE. We appreciate your time, consideration and further feedback you may provide. 

Sincerely,

Ron Ben-Amotz, Carly Sullivan, Joshua Zuckerman, Daniel James, Karl Maritato, Emily Morrison, and Riccarda Schuenemann

---

## [Editor Report · Decision Letter 1]

29 Apr 2022

Surgical and medical management in the treatment of proximal tibial metaphyseal fracture in immature dogs

PONE-D-21-29374R1

Dear Dr. Ben-Amotz,

We’re pleased to inform you that your manuscript has been judged scientifically suitable for publication and will be formally accepted for publication once it meets all outstanding technical requirements.

Kind regards,

Jason Organ

Academic Editor

PLOS ONE
---

## [Editor Report · Acceptance letter]

23 May 2022

PONE-D-21-29374R1 

Surgical and medical management in the treatment of proximal tibial metaphyseal fracture in immature dogs 

Dear Dr. Ben-Amotz:

I'm pleased to inform you that your manuscript has been deemed suitable for publication in PLOS ONE. Congratulations! Your manuscript is now with our production department. 

Kind regards, 

on behalf of

Dr. Jason Organ 

Academic Editor

PLOS ONE